# An Updated Review of the Biomarkers of Response to Immune Checkpoint Inhibitors in Merkel Cell Carcinoma: Merkel Cell Carcinoma and Immunotherapy

**DOI:** 10.3390/cancers15205084

**Published:** 2023-10-20

**Authors:** Adnan Fojnica, Kenana Ljuca, Saghir Akhtar, Zoran Gatalica, Semir Vranic

**Affiliations:** 1Institute of Virology, TUM School of Medicine, Technical University of Munich, 81675 Munich, Germany; adnan.fojnica@tum.de; 2Molecular Biology and Biochemistry, Gottfried Schatz Research Center, Medical University of Graz, 8036 Graz, Austria; 3Health Center of Sarajevo Canton, 71000 Sarajevo, Bosnia and Herzegovina; kenana.ljuca.medf@gmail.com; 4Department of Basic Medical Sciences, College of Medicine, QU Health, Qatar University, Doha 2713, Qatar; s.akhtar@qu.edu.qa; 5Department of Pathology, University of Oklahoma Health Sciences Center, Oklahoma City, OK 73019, USA; gatalicaz@gmail.com; 6Reference Medicine, Phoenix, AZ 85040, USA

**Keywords:** skin, Merkel cell carcinoma, therapy, immune checkpoint inhibitors, biomarkers

## Abstract

**Simple Summary:**

Merkel cell carcinoma (MCC) is a rare and highly aggressive type of skin neuroendocrine cancer that frequently recurs and metastasizes within a relatively short period. Despite rapid growth and characteristic skin color, MCC often goes undiagnosed in its early stage. Therefore, therapy is often initiated at the advanced stage, and selecting appropriate therapeutic interventions is critical. The emergence of novel immunotherapeutic agents, such as immune checkpoint inhibitors (ICI), presents a promising treatment option for advanced MCC. Several biomarkers, such as PD-L1 expression, tumor mutational burden (TMB), and microsatellite instability (MSI), showed significant potential as predictive biomarkers for treatment with ICI. Despite their predictive value, each has demonstrated limited value in MCC over recent years.

**Abstract:**

Merkel cell carcinoma (MCC) is primarily a disease of the elderly Caucasian, with most cases occurring in individuals over 50. Immune checkpoint inhibitors (ICI) treatment has shown promising results in MCC patients. Although ~34% of MCC patients are expected to exhibit at least one of the predictive biomarkers (PD-L1, high tumor mutational burden/TMB-H/, and microsatellite instability), their clinical significance in MCC is not fully understood. PD-L1 expression has been variably described in MCC, but its predictive value has not been established yet. Our literature survey indicates conflicting results regarding the predictive value of TMB in ICI therapy for MCC. Avelumab therapy has shown promising results in Merkel cell polyomavirus (MCPyV)-negative MCC patients with TMB-H, while pembrolizumab therapy has shown better response in patients with low TMB. A study evaluating neoadjuvant nivolumab therapy found no significant difference in treatment response between the tumor etiologies and TMB levels. In addition to ICI therapy, other treatments that induce apoptosis, such as milademetan, have demonstrated positive responses in MCPyV-positive MCC, with few somatic mutations and wild-type *TP53*. This review summarizes current knowledge and discusses emerging and potentially predictive biomarkers for MCC therapy with ICI.

## 1. Introduction

Merkel cell carcinoma (MCC) is a rare and highly aggressive type of skin neuroendocrine cancer that frequently recurs and metastasizes within a relatively short period [1,2,3]. MCC develops from Merkel cells, specialized cells located in the basal layer of the epidermis. These cells are involved in tactile sensation and play a critical role in mechanoreception [2]. MCC has a high mortality rate (30%) and is the second most common cause of skin cancer-related deaths after melanoma [2,3,4,5,6].

The prevalence of MCC varies among ethnic groups and geographic areas [6,7,8,9,10]. Although MCC has been observed in all races and ethnic groups, Caucasians have approximately a 25 times greater incidence rate than other groups [9,11]. MCC is primarily a disease of the elderly, with most cases occurring in individuals over 50 [1,9]. The global incidence of MCC is about 1.6 cases per 100,000 people per year, while the highest incidence rate is observed in Australia (3.9 cases per 100,000 men and 1.5 cases per 100,000 women) [9,10]. Norway has the lowest incidence rates of MCC, with only 0.45 cases per 100,000 men and 0.22 cases per 100,000 women [9,10]. The reasons for these variations are not fully understood and require further study.

In its early stage, MCC often goes undiagnosed despite its rapid growth and characteristic red/violet skin color [12], resulting in an average time between skin lesion detection and biopsy of approximately three months. By then, the average size of the tumors diagnosed at the biopsy is ~1.8 cm [12,13], potentially impacting the success of early treatment options [13,14]. The most common presenting symptom is a painless, rapidly growing bump, often located on the body’s sun-exposed areas (head and neck) [13,14]. According to Heath et al., over 50% of MCC lesions are initially clinically misdiagnosed as benign, with a cyst/acneiform lesion being the most frequent misdiagnosis (32%) [13]. Given its non-specific/misleading clinical features, a high level of suspicion is required for diagnosis [13], which is made by histopathologic examination of a biopsy, with immunohistochemical staining providing additional diagnostic clarity [15,16]. The biopsy shows solid nodular lesions in the dermis and subcutis consisting of the proliferation of small, round, and blue undifferentiated cells with high mitotic rate, apoptotic bodies, and occasional necrosis [15], as shown in Figure 1.

Immunohistochemically, the diagnosis of MCC is aided by the characteristic expression of cytokeratin 20 (CK20) and neuroendocrine markers, synaptophysin and chromogranin-A. MCC is typically negative for melanoma markers (S-100, Melan-A, HMB-45), lymphomas (LCA and related lymphoid lineage markers), adnexal carcinomas (CEA, EMA, CK7), and small-cell lung cancer [5,15]. It is essential to differentiate MCC from metastatic small-cell carcinoma (mSCLC) as they share some histopathologic and immunohistochemical features. MCC is considered the cutaneous equivalent of SCLC, and in some instances, patients with MCC may undergo additional screening for SCLC. In clinical practice, a diagnosis of MCC is based on a histopathological report that shows positive immunostaining for CK20 and negative thyroid transcription factor-1 (TTF-1) expression [15,16]. This is considered sufficient for diagnosing MCC in the absence of clinical signs of lung cancer, as these markers confirm the characteristic histopathological features of the disease [15,16]. However, other adjunct markers, such as MASH1/HASH1, may be used in the diagnosis as they are exclusively expressed in the SCLC [5].

The development of MCC involves a complex interplay between genetic, viral, and environmental stimuli such as ultraviolet (UV) exposure [7,17]. Most MCC cases are associated with the Merkel cell polyomavirus (MCPyV) infection, the key player in the development of MCC. The virus integrates into the genome of the host cells and disrupts the normal cell cycle control mechanisms, leading to cancer development [1,2,9]. MCPyV is present in approximately 80% of MCC patients (Figure 1). Its presence has been linked to a more indolent clinical course and fewer mutations in tumor cells than MCPyV-negative tumors (Figure 1) [2,9]. Exposure to UV radiation is another significant risk factor contributing to the development of skin cancer, including MCC, due to inducing DNA damage to the skin [1,2,8,9,18]. This damage can lead to mutations that disrupt normal cell cycle control mechanisms and promote the growth of cancerous cells [8,9]. No differences in clinical presentation between MCPyV-negative and MCPyV-positive tumors were observed [2,8,9].

The selection of therapeutic interventions for MCC is influenced by several critical factors, including the tumor stage, location, the extent of lymph node involvement, and the patient’s overall health status [10,11]. Surgery and radiotherapy are the standard modalities for achieving local disease control [11,19]. Chemotherapy is typically reserved for treating advanced/metastatic MCC, with limited efficacy [11,19]. Several chemotherapy regimens have been used over the years, among which the most common one is platinum-based drugs plus etoposide (PE) [20]. The overall response rate to PE is reported at 60%. Other standard chemotherapeutic options include the combination of cyclophosphamide, doxorubicin, and vincristine (CAV). CAV has a higher response rate (76%); however, significant side effects, including a death rate of 3.5%, are reported [20].

The emergence of novel immunotherapeutic agents, such as immune checkpoint inhibitors (ICIs), presents a promising treatment option for advanced MCC. However, the development of standardized treatments remains a subject of ongoing research [12,19,21,22].

The development of ICI therapy, such as those targeting programmed cell death ligand 1 (PD-L1) and anti-programmed death-1 (anti-PD-1), has significantly improved the treatment outcomes for patients with MCC [4,16,23]. However, not all MCC patients respond equally to ICIs; therefore, identifying biomarkers that predict responsiveness is an active area of research [1,4]. Similar to other cancers, several potentially predictive biomarkers have been identified and explored for MCC, including PD-L1 expression, tumor mutational burden (TMB), and the presence of microsatellite instability (MSI) [19,24,25,26,27,28]. Identifying and understanding these biomarkers may help select patients who would benefit from ICI therapies and adjust treatment strategies to improve disease outcomes.

## 2. Immunotherapies for the Merkel Cell Carcinoma

Currently, there are three approved ICI treatments by the Food and Drug Administration (FDA): one anti-PD-L1 monoclonal antibody treatment (avelumab) and two anti-PD-1 monoclonal antibody treatments (pembrolizumab and retifanlimab-dlwr) (Table 1) [29,30,31,32,33,34].

In May 2017, the FDA approved avelumab as a treatment for metastatic MCC in adults and children older than 12. Approval was based on phase two, a multicenter clinical trial involving 88 patients with advanced chemotherapy-resistant disease [29]. The actual trial started in 2014, intending to examine the effect of avelumab on patients with MCC who failed the first line of treatment (chemotherapy) [29]. All the samples were histologically confirmed as MCC [29]. Patients received therapies intravenously at a dose of 10 mg/kg every two weeks, and the primary endpoint was to achieve a high objective response rate (ORR) [29]. As of 26 September 2017, 88 patients were monitored over a median follow-up period of 29.2 months, ranging from 24.8 to 38.1 months. Of the 88 patients enrolled, 31.8% of them demonstrated an ORR. Within this response rate, 11.4% of patients achieved a complete response. Notably, 19 out of 29 patients with responses experienced ongoing benefits, including 12 patients with responses exceeding two years in duration [35]. The median duration of response (DOR) had not been reached, ranging from 2.8 to 31.8 months [29]. Treatment-related adverse events occurred in 5% of patients; no treatment-related grade four adverse events or treatment-related deaths were reported [29,31]. The follow-up study in the following year evaluated the efficacy of avelumab [31]. All patients from the previous studies were included [31], and ORR increased to 33.0%. The estimated proportion of responders with ≥1-year duration of response was 74%, while the estimated 1-year progression-free survival (PFS) rate was 30%, and the 1-year overall survival (OS) rate was 52%. These findings suggest some long-term benefits in a proportion of patients with previously treated MCC. Similarly, the trial that led to the pembrolizumab approval in 2018 enrolled 50 patients with advanced MCC [30]. The overall tumor response rate was 56%, the same response rate reported in a smaller-scale study also conducted in 2016 [30]. Among the patients with responses, more than half had a response lasting more than a year [30]. Seven patients discontinued their treatment due to experiencing side effects [30]. In 2020, the safety profile of avelumab was consistent with previous reports, while the ORR was 33% [36]. In the last update from 2022, avelumab monotherapy as a first-line treatment for patients with metastatic MCC (mMCC) demonstrated a noteworthy 4-year OS rate of 38%. These OS rates exceeded those observed in previous historical studies of first-line chemotherapy. These findings provide additional strong evidence for considering avelumab as the standard-of-care treatment for mMCC patients [35].

In March 2023, the FDA approved a new PD-1-blocking monoclonal antibody, retifanlimab-dlwr [23,32], for treating MCC based on the phase two clinical trial results, POD1UM-201 [23]. The study was a single-arm, open-label, multicenter endeavor that aimed to evaluate the safety and efficacy of retifanlimab-dlwr in patients with metastatic or recurrent locally advanced MCC [23]. Eighty-seven adult patients were involved in the study, and retifanlimab-dlwr was administered at 500 mg every four weeks for up to two years [23]. The primary efficacy analysis was based on 65 patients, and the ORR was 50.8%, with a complete response rate of 13.8% (Grignani et al., 2021). Of the patients who responded to treatment, 76% had a duration of response (DOR) of six months or longer, and 62% had a DOR of 12 months or longer. The safety population comprised 105 patients with MCC [23,32]. The most common adverse reactions reported were fatigue and musculoskeletal pain, while some patients experienced diarrhea, rash, and nausea [23,32].

In 2022, the FDA approved nivolumab, another antibody-based inhibitor of PD-1 for adult patients with resectable non-small-cell lung cancer (NSCLC). The approval was based on the CHECKMATE-816 trial. Some recent studies have examined the effect of nivolumab in MCC patients as well.

A recent study [33] reported a randomized, open-label, phase two trial that assessed treatment with combined nivolumab plus ipilimumab for MCC patients with advanced MCC [33]. The study found that first-line combined nivolumab and ipilimumab showed an ORR with durable responses and an expected safety profile [33]. Combined nivolumab and ipilimumab [a monoclonal antibody that blocks cytotoxic T lymphocyte antigen-4 (CTLA-4)] also showed clinical benefit in patients previously treated with anti-PD-1 and PD-L1 therapies [33].

This review will focus on the status of the current predictive biomarkers (PD-L1 expression, TMB, and MSI) to the ICI therapies in MCC.

### 2.1. Literature Search

The PubMed/MEDLINE/PubMed Central database was searched without specific filters for a general understanding of the topic. Scientific terms such as “Merkel cell carcinoma”, “tumor mutational burden/load”, “microsatellite instability/MSI/”, “biomarkers”, “immunotherapy”, “cancer”, “tumors”, and “immune checkpoint inhibitors” were searched.

However, for the summary and the critical review of the recent advancements, a literature search was limited to the articles published in the last five years (until February 2023). Specifically, for the case of the TMB marker, keywords “tumor mutation burden” or “tumor mutation load” or “TMB” in combination with “Merkel cell carcinoma” or “Merkel Polyoma Virus” or “MCC” were used. In the case of MSI, keywords such as “Microsatellite instability” or “Microsatellite instability-high” or “MSI” or “MSI-H” or “Mismatch repair” or “MMR” in combination with “Merkel cell carcinoma” or “Merkel Polyoma Virus” or “MCC” were used. For the PD-L1, keywords “Programmed Death-Ligand 1” or “PD-L1” combined with “Merkel cell carcinoma” or “Merkel Polyoma Virus” or “MCC” were used.

### 2.2. PD-L1 Status in Merkel Cell Carcinoma

PD-L1 is a transmembrane protein expressed (aberrantly) in various neoplastic cells or the immune cells of the tumor stroma [34]. More than two decades ago [35], this protein was recognized as an inhibitory ligand of PD-1 and expressed mainly on the surface of T cells, B cells, and natural killer (NK) cells [27,28,37,38]. Their binding leads to the suppression of T cells, thereby preventing our immune system from attacking cancer cells [28,38].

As PD-L1 expression in a tumor facilitates immune evasion, targeting (blocking) this immune checkpoint could enhance anti-tumor immunity and eliminate cancer cells [34,39]. Usually, the presence of PD-L1 is assessed with immunohistochemistry (IHC), and this biomarker is widely used and validated for predicting the response to ICI therapies in various cancer types [34,39,40,41].

Although PD-L1 expression on tumor cells has been utilized in numerous clinical trials and has approved clinical indications as a potential predictive biomarker for ICI response, the accuracy and reliability of FDA-approved PD-L1 expression assays and the application of PD-L1 as a predictive marker have raised many concerns [39,41]. Some patients who tested positive for PD-L1 expression may not respond to the corresponding ICI therapy. Conversely, patients who test negative for PD-L1 expression may still respond positively to the treatment [40]. Also, there are many concerns regarding the specificity of various anti-human PD-L1 antibodies used during IHC and the potential impact of tissue fixation and antigen retrieval techniques on assay results [34,39,41]. Although standardization of IHC assays has partially resolved some of these concerns, there is currently a lack of consensus on the appropriate threshold for defining PD-L1 positivity [34,39,41]. Usually, FDA-approved assays define PD-L1 positivity as having ≥5% of tumor cells exhibiting PD-L1 staining [42,43].

Over the years, it has been consistently observed that tumors with higher expression of PD-L1 tend to exhibit better response rates to ICI therapies [26]. Almost as a standard now, PD-L1 expression is considered high if a transmembrane protein is present in at least 50% of cancer cells [26]. Regarding MCC subtypes, PD-L1 expression in cancer cells is observed in both MCPyV-negative and MCPyV-positive MCC patients [44]. However, it is often the case that PD-L1 is more frequent in MCPyV-positive tumors than in MCPyV-negative MCC [44]. Furthermore, PD-L1 expression is often absent in the MCPyV-negative MCC [37,45].

Several studies have reported higher overall survival, MCC-specific survival and progression-free survival in patients with PD-L1+ tumors and intratumoral infiltration with CD8+ and FoxP3+ lymphocytes [44,45,46]. These results support the idea of blocking the PD-L1 signaling pathway as a new direction in the immunotherapy of MCC [44].

On the other hand, a study by Hanna GL et al. detected PD-L1 (clone 73–10) (a rabbit monoclonal recombinant antibody was used for the characterization) in about a quarter of MCC tumor cells, and PD-L1 in more than 90% of immune cells did not find a significant difference in overall survival and prognosis in patients who were not treated with ICI, including pembrolizumab, avelumab, and nivolumab [47].

Studies have come to several conclusions comparing MCC tumors of unknown primary origin to MCC tumors of primary cutaneous origin based on Merkel cell polyomavirus positivity. Regardless of MCC polyomavirus positivity, MCC tumors of unknown primary origin expressed higher PD-L1, and CD8+ and FOXP3+ infiltration, than MCPyV-positive primary cutaneous tumors. Furthermore, regardless of origin, MCPyV-negative MCC tumors showed higher TMB [44].

Hence, MCPyV-positivity is estimated to be a valid prognostic factor as it stimulates the immune system’s reaction [44,45].

Another study has found a correlation between the viral load of MCPyV and poor overall survival [48]. An interesting finding was the increased prevalence of MCC in women, which has not been fully clarified. Tumor site (head and arms region) and stromal infiltration with CD8+ lymphocytes correlated positively with PD-L1 status. However, PD-L1 expression did not affect the outcome [48]. This study suggests a local immune response was triggered by the virus, since a correlation between CD8+ infiltration, PD-L1 expression, and viral load was not found [48]. In another study, increased PD-L1 expression on immune and tumor cells correlated with the MCPyV+ status and was a favorable prognostic factor in non-metastatic disease [48].

The study by Acikalin et al. exploring the status of the EZH2 gene suggests that EZH2 could be a potential target in treating MCC. This gene codes the synthesis of the enzyme histone methyltransferase [49]. Via methylation of histones in chromosomes, this enzyme regulates and modifies the activity of specific genes which play a pivotal role in the etiology of different conditions, including MCC. This study found controversial effects of EZH2 expression in MCC, meaning that higher EZH2 expression was associated with metastasis or recurrence, whereas low EZH2 expression correlated with shorter overall survival [49].

Another study on MCC prognostic factors reported that virus negativity correlates with a higher prevalence of ulcerations, high neutrophil/CD8+ ratio, and E-cadherin downregulation, which are considered to have a negative prognostic value since they induce a tumor-promoting microenvironment by suppressing the infiltration of the CD8+ lymphocytes [50]. The study evaluated PD-L1 expression and tumor-infiltrating lymphocytes based on MCPyV positivity. They found that MCPyV infection affects the immunogenicity of MCC through high PD-L1 signaling and dense tumor-infiltrating lymphocytes [50].

Additionally, MCPyV+ tumors are protected by the layer of PD-L1/CD33+ cells on the periphery. On the other hand, MCPyV-negative tumors are predominantly PD-L1-negative [51].

Despite the high mortality rate and poor prognosis of MCC, advances in treating MCC through immunotherapy represent a promising direction for these patients. Patients who received immunotherapy responded positively to PD-1/PD-L1 blockade with atezolizumab, durvalumab, and avelumab, and they had a better prognosis [52].

Chemotherapy was the standard of care for patients with MCC until anti-PD-L1 antibody avelumab was approved in 2017 by the FDA and European Medicines Agency (EMA) [36,53,54,55].

The study by D’Angelo et al. investigated long-term survival outcomes in MCC patients and found that avelumab had an overall response rate of 33%, with a complete response rate of 11.4% of patients [36]. Additionally, over 80% of long-term survivors were patients with PD-L1+ tumors. These results imply that avelumab may be an effective treatment for MCC patients, particularly those with PD-L1 expression [36]. However, another study on ICI did not find a correlation between PD-L1 positivity and MCPyV status [56]. Complete response was reported in 44% of the patients, with a median time to respond of 8 weeks. MCPyV-negative tumors had a significantly higher objective response (69%) than MCPyV-positive tumors (43%) [56].

A study by Topalian et al. that considered neoadjuvant therapy with nivolumab in patients with resectable MCC, regardless of MCPyV, PD-L1 and TMB status, showed that 47% of patients treated with nivolumab and underwent surgery had a complete response, while 54% of patients had a tumor reduction of ≥30% [27].

However, although MCCs are divided into two groups (TMB-high/UV-driven and TMB-low/MCPyV-positive), a study by Knepper et al. in 2019 showed similar response rates to ICI in both tumor subtypes. Moreover, PD-1 instead of PD-L1 expression affected the response rate (77% vs. 21%) [40]. The most relevant studies on PD-L1 and immunotherapies are summarized in Table 2.

### 2.3. TMB Status in Merkel Cell Carcinoma

The variability in patient response to ICIs based on PD-L1 status reflects its numerous limitations [58,59,60] and underscores the need for additional biomarkers to improve the predictability of ICIs. Consequently, other biomarkers, including TMB and MSI, are utilized to select patients who may benefit from ICI therapies [58,61,62].

TMB accounts for the number of non-synonymous mutations in tumor DNA [21,63,64,65,66,67]. There is no universal consensus on the definitions for low and high TMB (TMB-L and TMB-H); TMB-L is considered ≤5 mutations per megabase (mut/Mb), while TMB-H is typically defined as ≤20 mut/Mb [64,68,69,70], with an intermediate TMB category defined as >5 and <20 mut/Mb [64,65,68,69,70,71,72,73,74]. FDA approval of pembrolizumab was based on the clinical trial in which TMB-H was ≥10 mut/Mb and was shown to be a good predictor of the therapy response [73].

The distinction between MCPyV-positive and MCPyV-negative MCCs may have important clinical implications. In most cases, IHC detection of the presence and absence of the MCPyV is based on the monoclonal antibody CM2B4, which is specific for the MCPyV large T antigen protein [75]. Other methods, such as PCR, are commonly used to diagnose MCPyV [76].

TMB seems to be a promising biomarker for predicting the effectiveness of ICI therapy in a subset of cancer patients [66,68,77,78]. Growing evidence supports this idea, particularly studies related to cancers with high levels of mutations, such as lung cancer and melanoma [65,67,78,79]. Most studies indicate that tumors with TMB-H tend to respond better to ICI treatment [65,66,67,77,78]. This assumption is based on the fact that a high number of mutations in a tumor leads to the formation of antigenic peptides, which further enhance the immunogenicity of the tumor and result in a better response to ICI [80,81,82,83,84]. Accordingly, patients carrying mutations that do not trigger immunogenic responses will have limited success from the ICI therapies.

MCPyV-positive MCCs are more common tumor types than MCPyV-negative MCCs, accounting for approximately 80% of cases and typically having a better prognosis [70,85]. The two types are strikingly different in their overall mutational characteristics, occasionally with 100-fold higher TMB in UV-induced versus MCPyV-positive MCC (Kaufman et al., 2018). The better prognosis of MCPyV-positive MCCs is often associated with distinctive molecular signatures [70,85]. MCPyV-positive MCCs are characterized by the clonal integration of the viral genome and the expression of viral oncoproteins, such as small T antigen (sT) [86]. sT has been identified as a critical driver of tumorigenesis in MCC. It leads to the inactivation of multiple tumor suppressor genes, resulting in cell cycle deregulation and genomic instability [86]. TMB-L is a potential marker often correlating with better patient outcomes for several tumor types [86]. TMB-L is commonly observed in MCPyV-positive MCCs; however, it is not uncommon for it to be reported in MCPyV-negative MCCs [76,85,86]. Harms and his colleagues reported a median TMB of 0.62 mut/Mb in MCPyV-positive MCC patients, significantly lower than the median TMB of 2.56 mut/Mb in MCPyV-negative MCCs [86]. Similarly, a study by Carter et al. identified that MCPyV-positive MCCs had a median TMB of 0.28 mut/Mb. In comparison, MCPyV-negative MCCs had a median TMB of 10.08 mut/Mb [87]. The TMB-L in MCPyV-positive MCCs may be because these tumors are driven by a viral oncogene, which limits the number of mutations that can accumulate in the tumor. In contrast to MCPyV-positive MCCs, MCPyV-negative MCCs are less common and generally associated with a worse prognosis [86]. This is supported by several studies that revealed poorer outcomes in patients with MCPyV-negative MCCs than those with MCPyV-positive MCCs [86,87,88]. These tumors are believed to be driven by UV-induced mutations and often have a TMB-H and a distinct mutational signature characterized by C > T transitions at dipyrimidine sites [25,86]. The TMB-H in MCPyV-negative MCCs may be due to their exposure to UV radiation, which induces DNA damage and leads to genomic instability. Despite these differences in molecular signature and prognosis, both MCPyV-positive and MCPyV-negative MCCs are treated similarly, with surgical excision being the primary treatment modality.

In the past five years, an increased effort has been made to investigate the correlation between TMB, ICI, and MCC [31,84,89,90,91,92,93,94,95,96]. The most relevant studies are summarized in Table 2.

The study from 2018 suggests that the TMB-H observed in Merkel cell carcinoma of unknown primary tumor (MCC-UP) patients may be one of the reasons for the improved outcomes seen in these patients, as compared with tumors with Merkel cell carcinoma of known primary tumor (MCC-KP) patients [95]. The group hypothesized that the TMB-H led to increased neoantigen presentation and immunogenicity compared to the tumors from MCC-KP patients [95]. Also, Donizy and his group [97] examined the IHC profiles of four groups of MCCs, including MCPyV-positive UP, MCPyV-negative UP, MCPyV-positive KP, and MCPyV-negative KP. They have identified distinct UV signatures in MCPyV-negative tumors and high immunogenicity in MCPyV-positive tumors [97].

In the study by Knepper et al. (2019), the differences between MCPyV-positive and MCPyV-negative MCC were investigated. Similar to the previous reports, TMB-L was found in MCPyV-positive MCCs, and MCPyV-positive MCCs were more likely to have mutations in the large T antigen gene [40]. TMB-H was observed for the MCPyV-negative MCC, and they were more likely to have mutations in other genes such as TP53, RB1, NOTCH1, and JAK1. Regarding ICI therapy, the study reported that patients with a TMB-H and higher expression of immune-related genes had a slightly better response to therapy, with a rate of 50% in the case of TMB-H and 41% for the TMB-L MCC tumors [40].

In another study from 2019, Gatalica et al. analyzed 48 MCC samples for the presence of MCPyV using IHC and correlated it with PD-L1, TMB and tumor mutational profiles. They found that 37.5% of the analyzed samples were MCPyV-positive. TMB was significantly lower in MCPyV-positive cases (6 mut/Mb) than in MCPyV-negative cases (25 mut/Mb) [98]. The most commonly mutated gene in MCPyV-negative cases was TP53 [98]. The group further suggested that avelumab therapy’s success in MCPyV-negative MCC cases may be related to the TMB-H [98].

A study conducted by Topalian et al. (2020) investigated patients with MCC using neoadjuvant nivolumab. Approximately half of the treated patients experienced pathological complete responses (pCRs) and radiographic tumor regressions, with nivolumab administered several weeks before surgery [27]. Also, a study has identified significantly higher levels of TMB in MCPyV-negative tumors compared with the MCPyV-positive ones [27]. Correspondingly, higher UV mutational signature scores were seen in MCPyV-negative versus MCPyV-positive tumors. However, the notable difference between TMB-L and TMB-H in achieving pCR was not evident [27].

Another study from 2020 reported the long-term data and biomarker analyses from the single-arm phase two JAVELIN Merkel 200 trial, in which the efficacy and safety of avelumab were examined for patients with MCC [36]. The safety profile of avelumab was consistent with previous reports, while the ORR was 33% [36]. Among the exploratory subgroups, ORR was highest (57.1%) for patients with MCPyV tumors that had TMB-H [36]. Median TMB was 2.72, non-synonymous with somatic variants (NSSV)/Mb for patients with MCPyV-negative, and 0.49 NSSVs/Mb for patients with MCPyV-positive tumors [36]. At the invasive margin, ORRs were also examined for other markers, such as PD-L1+ and CD8+ T cell density [36]. The median PFS was 3.7 months, while the median OS was 12.6 months [36]. In 2021, the same investigators reaffirmed better ORRs for patients with TMB-H and MCPyV-negative tumors.

Contrary to the avelumab therapy and better response observed in the MCPyV-negative tumors, clinical trials, in which pembrolizumab was applied as the first-line therapy for 50 MCC patients, had a different outcome [94]. Of the 50 MCC patients, the median ORR for pembrolizumab was 56–59% for MCPyV-positive patients and 53% for MCPyV-negative patients [94]. ORR was better for patients with TMB-L [94]. Similarly, another group examined an MDM2 inhibitor, milademetan, and its application for several MCC models [94]. Milademetan is a potent therapeutic agent suppressing cancer cell growth with wild-type (WT) TP53 [89]. Over the years, it has been observed that patients with MCPyV-positive MCC, having fewer somatic mutations, responded positively to the milademetan treatment [89]. This is most likely related to TP53, the most frequently mutated gene in MCC patients (54%), as MCPyV-positive tumors have few somatic mutations and usually express WT TP53 [89,99]. Inhibitors specifically bind to the MDM2 protein, preventing its interaction with TP53 and allowing p53 to become active. After activation, p53 continues to induce cell cycle arrest, apoptosis, and DNA repair. The use of MDM2 inhibitors seems to be a promising approach for treating MCPyV-positive and WT TP53 MCC [89].

The study by Horny et al. in 2021 aimed to investigate the mutational landscape of virus- and UV-associated MCC cell lines and compare it to the mutational landscape of MCC tumor samples [91]. They also performed whole-exome sequencing (WES) on four MCC cell lines. They compared the mutational landscape of the MCC cell lines with that of MCC tissue samples (27 MCPyV-positive and 38 MCPyV-negative MCC patients) [91]. Both MCPyV-associated and UV-associated MCCs were TMB-H [91]. MCPyV-negative MCC cell lines, on average, had 44.5 mut/Mb, while MCPyV-positive MCC cell lines had 10.5 mut/Mb. The study also reported that the mutational signature of UV-associated MCCs was dominated by C > T transitions. This single nucleotide variation (SNV) was also observed in 38% of MCPyV-positive MCC patients [91]. It is important to note that variations in the mutation rate may be caused by different methods [91].

Another research group studied 31 tumor samples to investigate copy number variants (CNV) in frequently altered genes [100]. They found deletions as the most common type of mutation, and no significant pattern was found between the two tumor subtypes (MCPyV-positive/negative tumors) [100]. However, an interesting observation related to the MCPyV-positive tumors was made, as these tumor types were capable of tumorigenesis, having only a few genomic mutations [100]. The researchers also attempted to identify a CNV pattern that could predict survival in MCC patients but did not observe any notable signatures [100].

A recent publication by Harms et al. (2021) expanded the MCC spectrum and further identified mutation profiles associated with the prognostic significance for MCPyV-positive MCC. The study identified a high incidence of TP53 and RB1 mutations in MCPyV-negative MCC, while a lower frequency of other mutations was identified for other genes [100]. TP53 and RB1 mutations were associated with worse prognosis [75]; however, this study indicated that prognosis should be based on MCPyV status rather than mutations in TP53 [75]. In a small series of MCCs, the study found that the activation of oncogenes was linked with a more aggressive disease progression in MCPyV-positive MCCs, as opposed to MCPyV-negative MCC [75].

Similarly, in the study by Brazel et al. (2023), actionable alterations annotated by the OncoKB database were associated with TMB. Of 313 patients, 82 had a TMB-H, while the rest had a TMB-L [24]. Further, the study examined the most common alterations for TMB-L and TMB-H. The most common mutations were observed in PIK3CA for TMB-H, while PTEN had the highest mutation rate for TMB-L cases [24].

Different approaches were used to understand the importance and role of TMB-H as a biomarker. What could be concluded from the multiple studies is that TMB-H has enormous potential as an immuno-oncology therapy biomarker. However, using TMB-H as the sole predictor of response to ICIs seems insufficient; accordingly, relying entirely on TMB to navigate ICI therapies would be insufficient. Even though, in most cases, TMB-H is associated with MCPyV-negative MCCs [27,40,75,100], there are occasions in which the contrary will be observed [76]. Also, TMB-H can be found in both MCPyV-positive and negative MCC etiologies [101]. Considering additional limitations with the cut-off values, the definition of TMB-H, and different sensitivity levels in the applied methods, additional studies and efforts are necessary to improve the potential of this biomarker. Ideally, other factors and biomarkers, such as specific mutation types, tumor microenvironment, PD-L1, and MSI, should be examined together with the TMB to get a more reliable indicator of responsiveness.

### 2.4. MSI Status in Merkel Cell Carcinoma

Another marker that emerges to supplement PD-L1 and TMB and tends to improve molecular prediction of ICI therapies is microsatellite instability (MSI) status [102,103].

Microsatellites (MS) are short (1–6 bp) and repetitive sequences of DNA that are scattered throughout the genome [104]. Alterations, including deletions and insertion in these microsatellites, are defined as MSI. MSI results from a functional deficiency in one or more major mismatch repair proteins (MMR), which correct such errors during DNA replication [104]. Therefore, MSI is a significant factor in the development and progression of tumors. The MMR deficiency is observed in numerous tumor types, and the most notable manifestation is in colorectal cancer, as each fifth patient will manifest the MSI phenotype [102,103,105]. MSI has also been detected in several other cancer types, including MCC [102,103,105].

PCR and NGS are the most common methods used to assess changes in repetitive sequences [102,106,107] directly. The panel of at least five markers (BAT25, BAT26, D2S123, D5S346, and D17S250) is a PCR-based test commonly used for this purpose, and MSI is defined as high (MSI-H) if two or more markers are unstable [108,109]. For the NGS approach, microsatellite loci in the target regions of the NGS panel are first identified using the MISA algorithm. Further analysis is applied to identify insertions and deletions that increase/decrease the number of repeats [102]. The assessment of the MMR protein expression is usually conducted by IHC, exploring the status of the four major MMR proteins MLH1, MSH2, MSH6, and PMS2, as reviewed in [23,32,57,109,110].

MSI-H cancers have great potential for enhanced responsiveness to anti-PD-1 therapies, such as pembrolizumab and nivolumab [40,94]. In the past, various malignancies, including colorectal, prostate and bladder cancer with MSI-H and deficient MMR, had positive responses once treated with the ICI [105,109,111]. The same effect was observed for the MSI-H neuroendocrine tumors [105,109,111]. Not many studies have examined the status of MMR and MSI in MCC; in particular, their potential as a predictor for ICI therapy success is not well understood [109].

In two studies from Caris Life Sciences [98,102], none of the 48 cases of MCC had MSI-H utilizing NGS, PCR, and/or IHC.

A study by Gambichler et al. investigated the expression of MMR proteins in MCC. They analyzed 56 MCC samples using IHC, and a significant association between low-expression MMR proteins (<10th percentile) and a negative MCPyV status was observed [109]. Nine patients had low-level MMR protein expression, while MSI evaluation was possible in five cases. Only one MCC sample was MSI-H [109]. Interestingly, PFS and the MCC recurrence rate did not significantly differ between low and high MMR samples [109]. When it came to therapy, most patients were taking conventional therapies, applied before ICI therapy [109]. Only nine patients with advanced stages of the disease received ICIs [109]. Of these nine patients, only one had a low-level expression of MMR proteins and was treated with avelumab [109]. Interestingly, the patient was MCPyV-negative and did not experience any MCC recurrence during a 40-month follow-up period [109]. The authors suggested additional trials to determine if the subset of MCC patients with low expression of MMR may respond better to ICI therapy.

Another recent study [71] performed comprehensive genomic profiling of MCC, with and without MCPyV integration. The study also examined the presence of TMB and MSI in 37 MCC samples. No single case of MSI-H in MCC samples was reported [71].

The most recent study, published in January 2023, has examined 324 samples and, to date, was the largest genomic analysis of MCC patients [24]. The group identified 20.2% of alterations as oncogenic, while MMR alterations were present in 8% of cases [24].

## 3. Emerging Predictive Biomarkers for ICI in Merkel Cell Carcinoma

Even though ICIs have substantially improved the outcome of multiple malignancies, response rates remain low. For MCC, the response rates to ICIs are around 50%. Primary and acquired resistance to ICI are commonly observed in MCC patients [112]. Long-term responses to the ICI are reported only for half of the responders [112]. The rationale for high ICI resistance is related primarily to the mutations of apoptosis-regulated genes such as TP53 and RB1 (especially in the case of MCPyV-negative MCC) but also due to the overexpression of anti-apoptotic proteins like Bcl-2 and Bcl-xl [113]. In Section 2.3, we already described that common mutations in MCC are related to TP53, RB1, NOTCH1, and JAK1 pathways [40]. The p53 protein regulates apoptosis in response to DNA damage and other cellular stresses [114]. A similar function of the Rb protein achieves this as well. Once the cell experiences DNA damage or adverse stimuli, the Rb protein promotes apoptosis through the inhibition of E2F-mediated transcription of pro-survival genes, enhanced expression of TP53, or modulation of Bcl-2 protein [115,116,117]. A good example of therapy resistance and TP53 mutations is the treatment of MCC cell lines with milademetan. Multiple MCC cell lines that contain wild-type TP53 demonstrated sensitivity to milademetan, even at nanomolar concentrations [89]. However, resistance to this drug was reported in the TP53 mutant cell lines such as MS-1 [89]. Another apoptosis-regulated mechanism is the expression of Bcl-2 and Bcl-xL. Merkel carcinoma cells express Bcl-2 and Bcl-xL constitutively [113,118]. In multiple MCC cell lines, high levels of Bcl-2 and Bcl-xL are reported, and downregulation of the proteins by RNA interferences promotes apoptosis [113,116]. Therefore, the modulation of the Bcl-2 and Bcl-xL expression should be an integral part of the ICI treatment strategy.

PD-L1, TMB, and MSI are established predictive biomarkers for ICI in different cancers. However, they are insufficient, as a substantial proportion of cancer patients may respond to ICIs without these biomarkers; similarly, their presence (expression) may not be associated with a favorable therapeutic response to ICI. Therefore, there is an unmet need to identify additional/new biomarkers and reassess the existing ones to improve the treatment response. Some of these emerging biomarkers include tumor-infiltrating lymphocytes (TILs), lactose dehydrogenase (LDH) levels, neutrophil/lymphocyte ratio (NLR), and circulating tumor DNA (ctDNA). None of these markers have been validated yet [57,110,119,120,121].

An example of the TIL application was observed in a study from 2020, where the clinical characteristics and potential biomarkers for 41 patients with MCC who received ICI treatment were examined [122]. Different biomarkers were analyzed, including TILs, serum LDH, NLR, PD-L1 expression, and MCPyV status [122]. Most notably, the prevalence of central memory T (TCM) with various T cell receptors among TILs was linked with a positive response to treatment [122].

Also, additional effort has been made to understand the prognostic and predictive potential of the NLR and LDH biomarkers in patients with MCC [123]. Torchio and his group examined an interesting case of a patient who had elevated NLR and LDH values prior to ICI treatment, followed by a rapid decrease in the levels of these markers after therapy [123]. An exceptionally complete response from the patient was reported. Interestingly, the MCC patient was PD-L1–negative and MCPyV-negative. In the sections above, we mentioned the patient’s case who responded better to the ICI therapy for the MCPyV-positive MCC. This indicates that other biomarkers should be included in selecting patients for ICI treatment besides MCPyV status.

ctDNA has emerged as a promising biomarker for prognostic and predictive applications in various cancer types, including MCC [120,121]. The non-invasive nature makes this biomarker attractive for monitoring disease progression and guiding treatment decisions [120,121]. Using ctDNA may help identify patients who may benefit from ICI therapies and maximize their application for MCC and other solid tumors. Thus, a recent case study used ctDNA to monitor the disease burden in a patient with MCC [121]. The level of ctDNA was wholly aligned with the levels of formed neoplasm. The highest ctDNA value observed was 42.45 mean tumor molecules [MTM]/mL after 42 days, when the neoplasm appeared most prominent [121]. After treatment with the pembrolizumab, no evidence of MCC recurrence was reported, and ctDNA had reduced to zero MTM/mL [121]. A personalized ctDNA assay demonstrated the predictive significance of this assay for an MCC patient treated with pembrolizumab [121]. Another study by Park et al. from 2022 involved the analysis of whole blood samples from 30 MCC patients, using WES of tumor, and matched normal blood to identify tumor-specific SNVs [120]. The study aimed to assess the utility of ctDNA as a biomarker for MCC and its potential use in monitoring disease burden and predicting treatment response [120]. Notably, elevated ctDNA levels in two patients led to early ICI therapy with rapid treatment responses, highlighting the clinical utility of ctDNA in managing MCC patients [120]. The future of ctDNA in MCC and other solid tumors looks promising. However, further interventional studies will be necessary to establish the use of ctDNA levels as a reliable tool to guide ICI treatment decisions.

## 4. Conclusions

ICI therapies have shown promising results in MCC patients. Although ~34% of MCC patients are expected to exhibit at least one of the predictive biomarkers (PD-L1, TMB, and MSI), their clinical significance in MCC is not fully understood. PD-L1 expression has been variably described in MCC, but its predictive value has not been established yet. Our literature survey indicates conflicting results regarding the predictive value of TMB in ICI therapy for MCC. Avelumab therapy has shown promising results in MCPyV-negative MCC patients with TMB-H, while pembrolizumab therapy has shown a better response in patients with TMB-L. A study evaluating neoadjuvant nivolumab therapy found no significant difference in treatment response between the tumor etiologies and TMB levels. In addition to ICI therapy, other treatments that induce apoptosis, such as milademetan, have demonstrated positive responses in MCPyV-positive MCC tumors with fewer somatic mutations. Additional clinical trials with a larger sample size would be necessary to obtain more conclusive results. Also, a combination of chemotherapy and other non-ICI therapies with the ICI should be considered in MCC. Previous studies have demonstrated success with combined treatments in several tumor types, including breast cancer.

Most studies from the last few years have reported that MCC is MSS. However, rare cases of successfully treated MSI-H MCC with ICI were reported. Although rare, they indicate a need for a comprehensive testing approach, including determining MMR status, because rare MCC patients may respond favorably to ICI therapies. Further research is required to better understand the extent and timing of ICI alone or in combination with other modalities. With the ongoing acceptance of ctDNA in disease monitoring, it can potentially improve approaches to managing MCC and other cancer types, offering personalized and targeted treatment options to all patients.

## Figures and Tables

**Figure 1 cancers-15-05084-f001:**
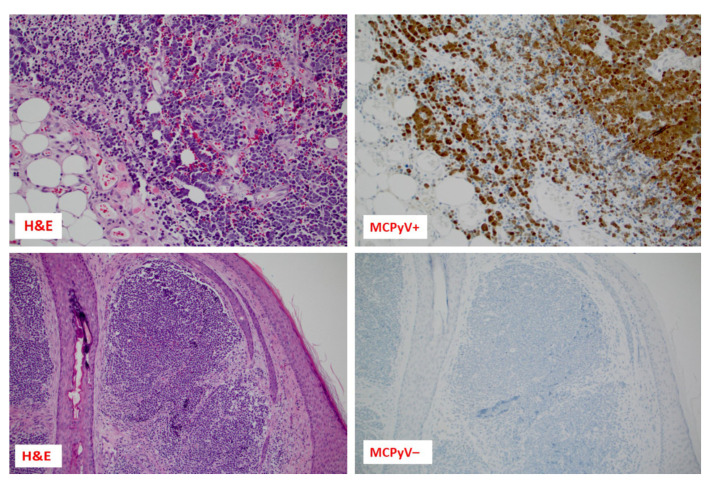
Morphology (Hematoxylin and Eosin stains/H&E/) of two cases of Merkel cell carcinoma with different etiology: upper images show a Merkel cell polyomavirus (MCPyV)-positive carcinoma, while lower images are a MCPyV-negative case; in both instances, MCPyV status was assessed using immunohistochemistry (anti-MCPyV large T antigen, clone CM2B4, Merck, Millipore, Burlington, MA, USA). Both cases had low PD-L1 expression by immunohistochemistry (≤5% of cancer cells; clones Ventana SP142 and Ventana SP263) and were microsatellite stable. However, their TMB status was different (the upper case had seven mut/Mb, while the TMB of the lower case was 34 mut/Mb).

**Table 1 cancers-15-05084-t001:** Approved ICI therapies for the treatment of Merkel cell carcinoma.

Immune Checkpoint Inhibitor (Year of Approval)	Mechanism of the ICIs	Reference
Avelumab (2017)	Avelumab functions by selectively targeting and blocking the PD-L1 protein.	[29]
Pembrolizumab (2018)	Pembrolizumab targets and blocks the PD-1 receptor.	[30]
Retifanlimab-dlwr (2023)	Retifanlimab-dlwr targets and blocks the PD-1 receptor.	[23]

**Table 2 cancers-15-05084-t002:** Overview of the most relevant recent studies regarding the treatment of Merkel cell carcinoma concerning TMB and PD-L1 status.

Clinical Studies	Number of Patients	Therapy	ORR (%)	mPFS (Months)	mOS (Months)
[29,31]	88	Chemotherapy/Avelumab	33%	30% (12)	52% (12)
[40]	57	ICI therapy	44%	NA	NA
[30]	50	Pembrolizumab	56%	67% (6)	NA
[36]	88	Avelumab	33%	21% (36)	31% (42)
[27]	39	Surgery/Nivolumab	47.2%	NA	79.4% (24)
[23]	65	Chemotherapy/Retifanlimab	50.8%	13.8% (7.4)	NA
[57]	114	Avelumab– Pembrolizumab/Nivolumab	47%	68% (12)	NA

NA—not available; mPFS—Median progression-free survival; mOS—Median overall survival; pCR—pathologic complete response.

## Data Availability

Not applicable.

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
