# Peer review of "An Updated Review of the Biomarkers of Response to Immune Checkpoint Inhibitors in Merkel Cell Carcinoma: Merkel Cell Carcinoma and Immunotherapy"

_cancers, 2023, doi:10.3390/cancers15205084_

Round 1

Reviewer 1 Report

The review article titled "An updated review of the biomarkers of response to immune 2 checkpoint inhibitors in Merkel cell carcinoma: Merkel cell              carcinoma and immunotherapy" by Fojnica et al., is a literature review on the use of immune check point inhibitors in Merkle Cell Carcinoma (MCC), with an emphasis on various mutational status and their implications on MCC treatment.  

The review article is developed nicely, adequate number of relevant references have been used. 

A major concern about this review is that I could not find any section that authors link  mutations to resistance to apoptosis. ICI allow immune cells to kill tumors via apoptosis, therefore, the authors should include a section about this critical issue. Authors are advised to highlight the additional sections for easier tracking if they decided to submit a revised version.

Moderate English editing is needed

Author Response

Thank you very much for your favorable review of the manuscript. You are absolutely right regarding the missing paragraph. Therefore, an additional section was added, and the link between mutations and resistance to apoptosis was explained (lines 556-577).

Additionally, we proofread the entire manuscript.

Reviewer 2 Report

The use of immune checkpoint inhibitors (ICI) as a treatment option for Merkel cell carcinoma (MCC) patients has yielded promising results. In this review, the authors aim to provide an up-to-date analysis of biomarkers associated with the response to immune checkpoint inhibitors in MCC. The review encompasses an overview of the existing literature, highlighting the limited number of studies in this particular field, and offers insightful recommendations for future research directions. While the premise of the review is intriguing, there is room for improvement in the organization and precision of the presented information to ensure that readers are not led astray.

1.     The JAVELIN Merkel 200 clinical trial is a well-documented study with subsequent follow-up investigations on patient outcomes. It is important to note, however, that the authors primarily emphasize data from the earliest phase of this trial. A more comprehensive analysis of the entire trial, including its subsequent phases and findings, could enhance the review's depth and accuracy.

2.     In the abstract, the authors mention the potential efficacy of milademetan in MCPyV-positive MCC cases with few somatic mutations. It is crucial to align this statement with the published data and cited references. If there is more recent or updated information regarding the performance of milademetan in MCPyV-positive MCC with few somatic mutations, it should be included in the review to maintain precision and credibility.

I don't have any specific comments

Author Response

Comments and Suggestions for Authors

The use of immune checkpoint inhibitors (ICI) as a treatment option for Merkel cell carcinoma (MCC) patients has yielded promising results. In this review, the authors aim to provide an up-to-date analysis of biomarkers associated with the response to immune checkpoint inhibitors in MCC. The review encompasses an overview of the existing literature, highlighting the limited number of studies in this particular field, and offers insightful recommendations for future research directions. While the premise of the review is intriguing, there is room for improvement in the organization and precision of the presented information to ensure that readers are not led astray.

  1. The JAVELIN Merkel 200 clinical trial is a well-documented study with subsequent follow-up investigations on patient outcomes. It is important to note, however, that the authors primarily emphasize data from the earliest phase of this trial. A more comprehensive analysis of the entire trial, including its subsequent phases and findings, could enhance the review's depth and accuracy.

Response to Reviewer#2: We agree with reviewer#2 that the JAVELIN Merkel 200 clinical trial is essential and well-documented. Therefore, we added more details on the set-up of the trial, subsequent phases and findings in order to enhance the review’s depth and accuracy. A complete description is found in lines 153-164.

  1. In the abstract, the authors mention the potential efficacy of milademetan in MCPyV-positive MCC cases with few somatic mutations. It is crucial to align this statement with the published data and cited references. If there is more recent or updated information regarding the performance of milademetan in MCPyV-positive MCC with few somatic mutations, it should be included in the review to maintain precision and credibility.

 Response: We rephrased the statement from the abstract from:

” In addition to ICI therapy, other treatments that induce apoptosis, such as milademetan, have demonstrated positive responses in MCPyV-positive MCC with few somatic mutation”. 

to

 “In addition to ICI therapy, other treatments that induce apoptosis, such as milademetan, have demonstrated positive responses in MCPyV-positive MCC with few somatic mutations and wild type TP53.”

Additionally, we provided a reference from 2022 that supports the statement that having MCPyV-positive MCC and only a few somatic mutations correlated with a positive response to therapy. The cited reference can be found in lines 445 to 450.

Additionally, we proofread the entire manuscript.

Round 2

Reviewer 1 Report

In response to this reviewer's previous critiques, the authors have revised the manuscript. The manuscript is suitable for publication now.

Editing is needed

Reviewer 2 Report

No further questions. 

No further questions.